# Data Mining to Select Relevant Variables Influencing External and Internal Workload of Elite Blind 5-a-Side Soccer

**DOI:** 10.3390/ijerph18063155

**Published:** 2021-03-18

**Authors:** José M. Gamonales, Kiko León, Daniel Rojas-Valverde, Braulio Sánchez-Ureña, Jesús Muñoz-Jiménez

**Affiliations:** 1Facultad Ciencias del Deporte, Universidad de Extremadura, 10005 Cáceres, Spain; fleon@unex.es; 2Centro de Investigación y Diagnóstico en Salud y Deporte (CIDISAD), Escuela Ciencias del Movimiento Humano y Calidad de Vida (CIEMHCAVI), Universidad Nacional, Heredia 86-3000, Costa Rica; 3Programa en Ciencias del Ejercicio y la Salud, Escuela en Ciencias del Movimiento Humano y Calidad de Vida (CIEMHCAVI), Universidad Nacional, Heredia 86-3000, Costa Rica; braulio.sanchez.urena@una.cr

**Keywords:** soccer, performance analysis, heart rate, technology, inertial measurement units

## Abstract

(1) Background: Data mining has turned essential when exploring a large amount of information in performance analysis in sports. This study aimed to select the most relevant variables influencing the external and internal load in top-elite 5-a-side soccer (Sa5) using a data mining model considering some contextual indicators as match result, body mass index (BMI), scoring rate and age. (2) Methods: A total of 50 top-elite visually impaired soccer players (age 30.86 ± 11.2 years, weight 77.64 ± 9.78 kg, height 178.48 ± 7.9 cm) were monitored using magnetic, angular and rate gyroscope (MARG) sensors during an international Sa5 congested fixture tournament.; (3) Results: Fifteen external and internal load variables were extracted from a total of 49 time-related and peak variables derived from the MARG sensors using a principal component analysis as the most used data mining technique. The principal component analysis (PCA) model explained 80% of total variance using seven principal components. In contrast, the first principal component of the match was defined by jumps, take off by 24.8% of the total variance. Blind players usually performed a higher number of accelerations per min when losing a match. Scoring players execute higher Distance_Explosive_ and Distance_21–24 km/h_. And the younger players presented higher HR_AVG_ and Acc_Max_. (4) Conclusions: The influence of some contextual variables on external and internal load during top elite Sa5 official matches should be addressed by coaches, athletes, and medical staff. The PCA seems to be a useful statistical technique to select those relevant variables representing the team’s external and internal load. Besides, as a data reduction method, PCA allows administrating individualized training loads considering those relevant variables defining team load behavior.

## 1. Introduction

Soccer 5-a-side (Sa5) is an adapted Paralympic sport for people with blindness [1]. The game has different modifications compared to soccer for persons without a disability. The game is played in a 40 × 20 m field with barriers on the sides and a ball that sounds when moving [2]. To be orienteered, players receive auditory references during the game by three sighted components; their coach, their goalkeeper, and the “caller”, who stands behind the opponent’s goal [3].

The Sa5 game requires some crucial adaptations of the sport technic and tactic because it is played in a total absence of visual references, which differentiates it from other soccer modalities. These unique characteristics are presented in scientific literature, with diverse approaches [4], highlighting performance analysis in sports, focusing on the main technical aspects [5,6], and defining performance indicators that explain the success in this sporting context [7,8,9,10,11].

In recent years, research on performance analysis in sports for people with disabilities has used new technological devices to quantify athletes’ external load. This load is usually linked with other internal load indicators (e.g., heart rate) to have a global perspective of the physical and physiological game demands [12,13,14]. The use of the so-called electronic performance and tracking systems (EPTS) has increased due to its portability and availability of many variables. These devices integrate multiple sensors (e.g., accelerometers, gyroscopes, magnetometers, global position systems and tracking technology) [15], allowing to record data related to the workload profile. These variables bring information about the magnitude and frequency of accelerometric-based (e.g., impacts, jumps, steps, landings, changes of direction) and speed-based indicators (e.g., distance, mean speed). Due to the sensor’s capacity, up to 250 variables are registered at a sampling frequency of 100 to 1000 Hz. This is why this information is defined as big data [16,17]. To identify, cluster, and select the most relevant variables in a specific team and task, it is necessary to implement a data mining technique [18]. The reduction in the data sets size allows to objectively identify the variables to assess the interaction between the external and internal load with other contextual factors that may influence performance.

One of these data mining methods is known as Principal Component Analysis (PCA). This is the most used technique to reduce data into a small series of variables (5–15 variables), explaining up to 70% of the data set variance. Although some studies have applied this statistical method to select and cluster anthropometric, biomechanical, physical, and physiological variables of team top elite sports, this procedure has never been used in the performance analysis of adapted sports for people with disabilities [19].

Considering that some contextual and situational factors may influence game’s physical and physiological performance, as other studies have revealed, it is hypothesized that due to the particular characteristics of the Sa5 game, the magnitude of some external and internal load variables could be influenced by contextual variables as match result, age, goal scorer and anthropometric variables. We hypothesized that both internal and external load variables may be affected by match result, age, goal scorer and some anthropometric variables, based on previous results of similar studies. Consequently, this study aimed to explore the potential differences in the Sa5 internal and external load variables selected by PCA considering three variables: a. match result, b. body mass index (BMI), c. scoring rate and d. age.

## 2. Materials and Methods

### 2.1. Study Design

A cross-sectional and observational protocol approached the problem of the study. The external and internal load variables were assessed during a Sa5 International Tournament. This congested tournament was held on a single weekend, and the teams were representatives of European national teams. A total of four matches were played per each team under similar circumstances and in the same venue. The Sa5 International Tournament was held in the city of Seville (Spain) in May 2019 (temperature = 21.2 ± 3.7 °C, relative humidity = 65.2 ± 5.3%), in which the teams of Spain, Italy, the Czech Republic, and an Andalusian team participated.

### 2.2. Participants

Data were collected from 50 blinded male players (age 30.86 ± 11.2 years, weight 77.64 ± 9.78 kg, height 178.48 ± 7.9 cm). The players were members of top elite national European teams. There were no reports of neuromuscular injuries that could compromise physical or physiological performance. The players who participated >85% of the matches’ effective time were included in the analysis.

Before data collection, all participants signed an informed consent document, which contained all the investigation details. The protocol followed the Helsinki Declaration guidelines for biomedical research. The study was reviewed and approved by the Institutional Review Board (University of Extremadura, Reg. Code 67/2017). Besides, the teams’ staff and tournament managers gave their consent for participation in this research.

### 2.3. Instruments and Procedures

Before the matches, each player was equipped with a heart rate monitor (GARMIN^TM^, Lenexa, KS, USA) to record internal load variables. Magnetic, angular rate, and gravity (MARG) sensors (WIMU^TM^, RealTrack Systems, Almería, Spain) were fixed using an adapted anatomically harness at the height of T2–T4 vertebrae between scapulae. The devices were calibrated considering previously published guidelines [20]. The MARG sensors integrate data of four three-axis microelectromechanical system accelerometers, gyroscope, and magnetometer to assess external load-related variables. After registration, the data was processed using special software (SPRO^TM^, RealTrack Systems, Almería, Spain) immediately after each match. Moreover, the MARG sensor incorporate GPS and UWB systems to register all speed and distance based variables. The sampling was made using a 5 Hz sampling frequency for the GPS system and 100 Hz for accelerometric-based variables [21]. All players were used to wear the MARG sensors during soccer matches.

### 2.4. Variables

While the MARG sensors provide up to 200 variables, considering there could be differences in the time played in each match, only peak and time-related variables were initially selected (n = 44). This criterion is used to homogenize the average demands of each match [22,23,24]. These variables were introduced in a correlation matrix to select the uncorrelated representative variables; those with an r > 0.7 were discarded [25]. Before an Exploratory Factor Analysis called Principal Component Analysis (PCA), this procedure was performed, and previous similar studies’ statistical protocol was followed [19,22]. Variables with variance = 0 were excluded from the analysis.

The 15 selected variables after the correlation matrix were scaled and centered (Z-scores). Suitability of PCA was confirmed by Kaiser-Meyer-Olkin (KMO = 0.71), and Barleth’s Sphericity test was significant (*p* < 0.01) [26,27]. Eigenvalues >1 were considered for the extraction of principal components (PC) [27]. A VariMax-orthogonal rotation method was performed to identify high correlations of components and guarantee that each principal component offered different information. A threshold of 0.6 in each PC loading was retained for interpretation, extracting the highest factor loading when a cross-loading was found between PCs. PCA procedure followed standard quality criteria, meeting 21 out of 21 of the quality items [19]. Moreover, the report of the PCA model results followed previous guidelines for team sport performance analysis [28].

The contextual variables considered during this tournament were match result (losers vs. winners), BMI (lower vs. higher based on the BMI mean), scoring rate (those scorers at least once during a match vs. non-scorers), and age (younger vs. older players based on the age mean).

### 2.5. Statistical Analysis

Data were expressed in mean and standard deviation with respective upper and lower limits for descriptive purposes. Independent *t*-tests were performed to explore differences by match result (loser vs. winner), BMI (lower vs. higher), scoring rate (scorer vs. non-scorer) and age (younger vs. older). The magnitude of *t*-test differences was interpreted using Cohen’s d (d) as follows: 0–0.2 trivial; 0.2–0.5 low; 0.5–0.8 moderate and >0.8 high [29]. Statistical differences were considered if *p* < 0.05. Data exploratory analysis was performed using the Statistical Package for the Social Sciences (SPSS, IBM, SPSS Statistics, v.22.0 Chicago, IL, USA).

## 3. Results

Table 1 shows absolute descriptive data of the 15 selected variables of the whole match. A total of seven PCs were extracted after performing PCA. The whole model explains 80.5% of the total variance. PC cumulative variance of each PC is shown in Table 1.

The results suggested differences by match result in only one external load variable. There were differences by match results in Acc per minute, and the losers make higher accelerations during the matches compared to the winners (see Table 2). This change was qualified as low based on effect size.

Additionally, there were no differences in external or internal workload demands considering BMI (see Table 3). No differences were found between those players categorized in low or high BMI. Besides, effect sizes showed trivial changes by group.

There were found differences in distance variables extracted from the PCA model. Those players who scored at least once during the matches showed higher Distance_Explosive_ and Distance_21–24 km/h_ than non-scorers (see Table 4). These variables showed low changes between groups based on the effect size.

There were found differences by age in HR_AVG_ and Acc_Max_ as internal and external load variables, respectively. Younger players performed higher HR_AVG_ and Acc_Max_ compared to the older ones. Effects sizes of these changes were classified as low (see Table 5).

## 4. Discussion

The purpose of this study was to explore the potential differences in the Sa5 internal and external load variables selected by PCA considering three variables as match result, body mass index (BMI), scoring rate and player’s age. The results suggested that players usually performed a higher number of accelerations per min when losing a match. There were no differences in external workload variables by player’s BMI. Those Sa5 players that scored during a match execute higher Distance_Explosive_ and Distance_21–24 km/h_. And younger players presented higher HR_AVG_ and Acc_Max_.

In actual soccer-playing, high-intensity actions (e.g., sprints, high-speed running, jumps, etc.) are considered the most relevant external load performance indicators in team sports. Moreover, these variables seem to be crucial during congested fixture conditions, and it is essential to understand how these external and internal load variables could be affected by some contextual factors [22]. Addressing these potential load differences could be fundamental when planning and monitoring both training and official matches.

In this research, 80.5% of the total variance was explained by the PCA model using seven PCs. The 15 variables extracted from the model were heart rate, accelerations, decelerations, high-intensity speed, impacts, jumps, and take-offs. The present PCA results are in line with analyses previously performed in team sports3/17/2021 10:34:00 PM were external and internal load variables derived from MARGS and inertial measurement units in team sports were selected through PCAs [19]. After the variables extraction, the results suggested some impact on external and internal loads by match outcome, scoring rate, and age.

It has been found that match outcome is significantly related to performance in some external and internal load variables in soccer. In this study, it was found that losers teams performed more accelerations per minute than the winners. In this sense, it has been also reported that total distance increases when losing even when distance performed between 0–6 km/h decreased [30]. These results suggest that high-intensity actions as accelerations could increase to improve the final score as the losing team is trying to look after new goal opportunities [30]. Although, there is a lack of agreement in soccer research since the link between the higher external and internal is not clearly understood; considering that other match contextual factors as opponent level, team strategies, or match status could also influence load and it is difficult to isolate the individual effect of these contextual variables [30,31,32]. The external and internal load demands could also increase and decrease during the match whether the team is winning, drawing, or losing [30].

While in soccer, BMI seems to be a crucial parameter that determines muscle power, fatigue index, and strength [33,34]; in Sa5, BMI does not influence external or internal load demands. BMI is a relevant indicator of performance that allows differentiation between players level and defines in-field actions as speed [35]. In Sa5, the developed speed during matches is lower than in regular similar soccer modalities [36]; it could be why BMI is not that determinant in the blinded player’s load. Speed could not be as critical as the change of direction, acceleration, or other external load variables as stated in recent studies [37,38]. Despite this evidence, more studies should be developed to explore the above more in-depth.

While the practice of any sport predisposes individuals to suffer injuries, particularly in Sa5 soccer, there are up to 80% of injuries of traumatic etiology [39]. Despite there are some rules that protect blinded players from an injury (e.g., the “go” rule), the inherent danger of hitting other players remains latent. As with other anthropometric variables, BMI should be monitored and controlled to enhance performance and prepare the body to resist those impacts.

In this study, those who scored at least once presented higher Distance_Explosive_ and Distance_21–24 km/h_. This may suggest the requirement to train the ability to perform high-intensity actions in these players. Considering that the actions usually started in the pre-offensive zone of the field had more probabilities of finish in goal in Sa5 [7], the attention should be focus on those players that due to tactical or positional characteristics play in this zone.

This result is also in line with other studies that suggest that most soccer goals are preceded by at least one high-intensity action performed by the assisting or scoring player [40]. Straight sprints, jumps, rotations and change of directions were the most common actions executed prior a goal situation. Moreover, decisive passes are preceded by these kind of high intensity actions enabling the attackers to locate in an optimal position to score [40].

In the present study, younger players presented higher HR_AVG_ and Acc_Max_ when compare to the older ones. In this respect, older players are usually present some physio-psychological factors as higher experience, higher self-confidence, better game reading and self-efficacy, allowing to dose their effort throughout the match [41] requiring less physical demands. Conversely, there are contrast evidence if the younger players present better physical performance in professional soccer compare to their older peers [42]. Besides it is clearly known that the rate of maturation impacts the physical performance characteristics of the players. Still, it usually depends on the relative age effect and no in broader age ranges as in professionals [42]. In this sense, national team coaches should consider these differences when identifying and selecting representative players.

## 5. Limitations

While the results of this study have provided information regarding how contextual factors as age, scoring abilities and match result could affect external and internal load demands, these research outcomes must be seen in the light of some limitations. One of the limitations in this study concerns the sample studied, the locomotion demands presented in this research must be applied to top-elite blinded soccer players and could serve as a reference to other inferior divisions. Due to the authors did not influence the natural dynamic of the competitions or the tournament schedule, the sample between the compared groups were distributed unequally. Besides, despite the matches were performed under the same conditions for all players (e.g., same venue, same turf, same referees) some other situational factors that could affect the game demands were not controlled (e.g., temperature). Finally, due to the lack of evidence in Sa5, some other soccer modalities were studied as reference.

## 6. Conclusions

This study represents the first effort to report external and internal load demands of Sa5 elite soccer, selecting, using an objective statistical criterion, those representative variables with great influence in physical performance. It was found that players usually performed higher number of accelerations per min when losing a match possible due to the need to increase their efforts to score. Those Sa5 players that scored during a match execute higher Distance_Explosive_ and Distance_21–24 km/h._ Considering the goals are usually anteceded by high-intensity actions (e.g., jumps, sprints), those Sa5 players that scored during a match execute higher Distance_Explosive_ and Distance_21–24 km/h._ Due to some technical and tactical Sa5 conditions that differentiate the more experienced players from the younger ones, the younger players presented higher HR_AVG_ and Acc_Max_.

## 7. Practical Applications

Coaches, athletes and medical staff should address the influence of some contextual variables on external and internal load during top elite Sa5 official matches. The comprehension of the potential differences by match outcome, players age and scoring rate could redirect the designing of conditioning training programs, match strategies, tactical decisions and recovery protocols, especially during congested fixture tournaments.

Coaches should acknowledge the match partial results to adequate tactical tasks, that may improve the team demands optimizing general performance. The coaches should also consider age differences between players to equilibrate the team demands to avoid fatigue effect and potential performance decrease. Besides, those players demanded to score should undergo particular high-intensity training tasks, considering that in soccer goals are usually preceded by peak accelerations, decelerations, high-intensity running and sprints.

Finally, PCA seems to be a useful statistical technique to objectively select those relevant variables that represent teams external and internal load. Data mining techniques may be applied to MARG sensors derived data to improve the data management and technical report to the coaching and medical staff. This data reduction method allows administrating individualized training loads considering those relevant variables defining team load behavior.

## Figures and Tables

**Table 1 ijerph-18-03155-t001:** External workload and locomotion variables extracted from principal components analysis in soccer 5-a-side playing.

	M ± DS	PC1	PC2	PC3	PC4	PC5	PC6	PC7
Eigenvalue	Whole Match	4.2	2.9	1.8	1.6	1.1	1.1	1
% Variance		24.8	1.3	10.3	9.3	6.5	6.4	5.9
% Cumulative Variance		24.8	42.1	52.4	61.7	68.2	74.6	80.5
Jumps (n/min)	0.08 ± 0.08 (0.01; 0.5)	0.99						
TakeOff_0–3 g_ (n/min)	0.07 ± 0.05 (0; 0.25)	0.86						
TakeOff_3–5 g_ (n/min)	0.01 ± 0.04 (0; 0.25)	0.85						
Acc (n/min)	42.41 ± 4.54 (33.88; 54.46)		−0.81					
HR_AVG_ (bpm)	138.53 ± 19.86 (87; 171)		0.84					
Distance_Explosive_ (m/min)	2.72 ± 2.03 (0; 8.42)			0.7				
Distance_21–24 km/h_ (m/min)	0.02 ± 0.11 (0; 0.69)			0.71				
Dec_5–4 m/s_ (n/min)	0.01 ± 0.02 (0; 0.09)			0.67				
Impacts_5–8 g_ (n/min)	1.74 ± 1.67 (0; 6.85)			0.61				
Acc_4–5 m/s_ (n/min)	0 ± 0.01 (0; 0.03)				0.83			
Dec_6–5 m/s_ (n/min)	0 ± 0.01 (0; 0.03)				0.87			
Acc_Max_ (m/s)	3.33 ± 0.99 (33.88, 54.46)					0.81		
Acc_6–10 m/s_ (n/min)	0 ± 0 (0; 0.02)					0.96		
Acc_5–6 m/s_ (n/min)	0 ± 0 (0; 0.02)						0.95	
TakeOff_5–8 g_ (n/min)	0 ± 0.01 (0; 0.03)							0.93

**Table 2 ijerph-18-03155-t002:** External workload and locomotion variables in soccer 5-a-side playing by match result.

Variable	Winner (n = 23)	Loser (n = 27)	t (*p* Value)	Cohen’s *d*
Jumps (n/min)	0.09 ± 0.05 (0.02; 0.22)	0.08 ± 0.09 (0.01; 0.5)	0.37 (0.71)	0.05
TakeOff_0–3 g_ (n/min)	0.08 ± 0.05 (0.02; 0.22)	0.07 ± 0.05 (0; 0.25)	1.02 (0.31)	0.14
TakeOff_3–5 g_ (n/min)	0.01 ± 0.02 (0; 0.08)	0.01 ± 0.05 (0; 0.25)	−0.64 (0.53)	0.09
Acc (n/min)	40.73 ± 3.91 (33.88; 51.14)	43.8 ± 4.61 (38.26; 54.46)	−2.55 (0.01) *	0.4
HR_AVG_ (bpm)	142.7 ± 3.91 (93;170)	134.85 ± 20.55 (87; 171)	1.39 (0.17)	0.2
Distance_Explosive_ (m/min)	2.64 ± 2.37 (0.07; 8.42)	2.79 ± 1.72 (0; 5.96)	−0.26 (0.8)	0.04
Distance_21–24 km/h_ (m/min)	0.05 ± 0.16 (0; 0.69)	0 ± 0 (0; 0)	1.52 (0.14)	0.21
Dec_5–4 m/s_ (n/min)	0.01 ± 0.02 (0; 0.09)	0.01 ± 0.01 (0; 0.04)	0.39 (0.18)	0.06
Impacts_5–8 g_ (n/min)	0.01 ± 0.02 (0; 0.08)	0.02 ± 0.04 (0; 0.17)	0.04 (0.97)	0.01
Acc_4–5 m/s_ (n/min)	0 ± 0.01 (0; 0.03)	0 ± 0.01 (0; 0.03)	0.27 (0.79)	0.04
Dec_6–5 m/s_ (n/min)	0 ± 0.01 (0; 0.02)	0 ± 0.01 (0; 0.03)	−0.9 (0.37)	0.13
Acc_Max_ (m/s)	3.2 ± 0.92 (1.43; 5.4)	3.44 ± 1.06 (1.07; 7.43)	−0.87 (0.39)	012
Acc_6–10 m/s_ (n/min)	0 ± 0.01 (0; 0.02)	0 ± 0.01 (0; 0.02)	−0.92 (0.36)	0.13
Acc_5–6 m/s_ (n/min)	0 ± 0.01 (0; 0.02)	0 ± 0.01 (0; 0.02)	1.09 (0.28)	0.15
TakeOff_5–8 g_ (n/min)	0 ± 0.01 (0; 0.03)	0 ± 0.01 (0; 0.01)	1.21 (0.23)	0.17

* Significant differences with low effect size.

**Table 3 ijerph-18-03155-t003:** External workload and locomotion variables in soccer 5-a-side playing by BMI.

Variable	Lower BMI (n = 34)	Higher BMI (n = 16)	t (*p* Value)	Cohen’s *d*
Jumps (n/min)	0.08 ± 0.05 (0.02; 0.22)	0.09 ± 0.11 (0.01; 0.5)	−0.56 (0.58)	0.08
TakeOff_0–3 g_ (n/min)	0.07 ± 0.04 (0.02; 0.22)	0.07 ± 0.06 (0; 0.25)	0.04 (0.97)	0.01
TakeOff_3–5 g_ (n/min)	0.01 ± 0.02 (0; 0.08)	0.02 ± 0.06 (0; 0.25)	−1.19 (0.24)	0.17
Acc (n/min)	41.71 ± 4.73 (33.88; 54)	43.9 ± 3.82 (39.46; 54.46)	−1.19 (0.24)	0.17
HR_AVG_ (bpm)	140.52 ± 19.15 (87; 170)	134.44 ± 21.3 (91; 171)	−1.75 (0.09)	0.25
Distance_Explosive_ (m/min)	2.9 ± 2.13 (0; 8.42)	2.33 ± 1.79 (0.24; 5.96)	0.97 (0.34)	0.14
Distance_21–24 km/h_ (m/min)	0.03 ± 0.13 (0; 0.69)	0 ± 0 (0; 0)	0.93 (0.36)	0.13
Dec_5–4 m/s_ (n/min)	0.01 ± 0.02 (0; 0.09)	0.01 ± 0.01 (0; 0.04)	0.95 (0.35)	0.14
Impacts_5–8 g_ (n/min)	1.68 ± 1.6 (0; 5.9)	1.85 ± 1.87 (0.1; 6.85)	0.97 (0.34)	0.14
Acc_4–5 m/s_ (n/min)	0.01 ± 0.01 (0; 0.03)	0 ± 0.01 (0; 0.02)	1.02 (0.32)	0.14
Dec_6–5 m/s_ (n/min)	0.01 ± 0.01 (0; 0.03)	0 ± 0 (0; 0.01)	1.18 (0.24)	0.14
Acc_Max_ (m/s)	3.26 ± 0.89 (1.07; 5.4)	3.47 ± 1.2 (2.28; 7.43)	−0.69 (0.5)	0.1
Acc_6–10 m/s_ (n/min)	0 ± 0.01 (0; 0.02)	0 ± 0.01 (0; 0.02)	−1 (0.33)	0.14
Acc_5–6 m/s_ (n/min)	0 ± 0 (0; 0.02)	0 ± 0.01 (0; 0.02)	−1.48 (0.15)	0.21
TakeOff_5–8 g_ (n/min)	0 ± 0.01 (0; 0.03)	0 ± 0 (0; 0.01)	0.64 (0.53)	0.09

**Table 4 ijerph-18-03155-t004:** External workload and locomotion variables in soccer 5-a-side playing by goal scorer.

Variable	Non Scorer (n = 32)	Scorer (n = 13)	t (*p* Value)	Cohen’s *d*
Jumps (n/min)	0.09 ± 0.09 (0.02; 0.5)	0.07 ± 0.04 (0.01; 0.17)	0.54 (0.59)	0.08
TakeOff_0–3 g_ (n/min)	0.07 ± 0.05 (0.02; 0.25)	0.07 ± 0.04 (0; 0.14)	0.18 (0.86)	0.03
TakeOff_3–5 g_ (n/min)	0.01 ± 0.05 (0; 0.25)	0 ± 0.01 (0; 0.04)	0.82 (0.42)	0.1
Acc (n/min)	42.68 ± 4.44 (33.88; 54.04)	41.64 ± 4.9 (34.58; 54.46)	0.71 (0.48)	0.1
HR_AVG_ (bpm)	138.56 ± 20.32 (87; 171)	138.46 ± 19.32 (91; 168)	0.01 (0.99)	0
Distance_Explosive_ (m/min)	2.36 ± 1.58 (0; 5.02)	3.74 ± 2.79 (0.24; 8.42)	−2.18 (0.03) *	0.31
Distance_21–24 km/h_ (m/min)	0 ± 0.01 (0; 0.02)	0.08 ± 0.21 (0; 0.69)	−2.44 (0.02) *	0.34
Dec_5–4 m/s_ (n/min)	0.01 ± 0.01 (0;0.04)	0.02 ± 0.03 (0; 0.09)	−1.34 (0.19)	0.19
Impacts_5–8 g_ (n/min)	1.53 ± 1.54 (0; 6.85)	2.33 ± 1.94 (0.1; 5.9)	−1.51 (0.14)	0.21
Acc_4–5 m/s_ (n/min)	0 ± 0.01 (0; 0.03)	0.01 ± 0.01 (0; 0.03)	−0.45 (0.65)	0.06
Dec_6–5 m/s_ (n/min)	0 ± 0.01 (0; 0.03)	0 ± 0.01 (0; 0.02)	−0.07 (0.95)	0
Acc_Max_ (m/s)	3.34 ± 1.1 (1.07; 7.43)	3.3 ± 0.64 (2.28; 4.29)	0.14 (0.89)	0.02
Acc_6–10 m/s_ (n/min)	0 ± 0 (0; 0.02)	0 ± 0 (0; 0)	0.59 (0.56)	0.08
Acc_5–6 m/s_ (n/min)	0 ± 0 (0; 0.02)	0 ± 0 (0; 0)	0.59 (0.56)	0.08
TakeOff_5–8 g_ (n/min)	0 ± 0.01 (0; 0.03)	0 ± 0.01 (0; 0.02)	0.32 (0.71)	0.05

* Significant differences with low effect size.

**Table 5 ijerph-18-03155-t005:** External workload and locomotion variables in soccer 5-a-side playing by age.

Variable	Younger (n = 22)	Older (n = 28)	t (*p* Value)	Cohen’s *d*
Jumps (n/min)	0.07 ± 0.03 (0.03; 0.13)	0.1 ± 0.1 (0.01; 0.5)	−1.06 (0.29)	0.15
TakeOff_0–3 g_ (n/min)	0.07 ± 0.03 (0.03; 0.11)	0.08 ± 0.06 (0; 0.25)	−0.54 (0.59)	0.08
TakeOff_3–5 g_ (n/min)	0 ± 0.01 (0; 0.02)	0.02 ± 0.05 (0; 0.25)	−1.26 (0.22)	0.18
Acc (n/min)	41.17 ± 4.47 (33.88; 54.04)	43.38 ± 4.43 (37.87; 54.46)	−1.74 (0.09)	025
HR_AVG_ (bpm)	147.17 ± 4.47 (33.88; 54.04)	131.04 ± 20.79 (87; 168)	3.19 (0.003) *	0.45
Distance_Explosive_ (m/min)	3.13 ± 1.49 (0.55; 6.44)	2.4 ± 2.34 (0; 8.42)	1.29 (0.2)	0.18
Distance_21–24 km/h_ (m/min)	0.03 ± 1.49 (0.55; 6.44)	0.01 ± 0.07 (0; 0.35)	0.58 (0.56)	0.08
Dec_5–4 m/s_ (n/min)	0.01 ± 0.01 (0; 0.04)	0.01 ± 0.02 (0; 0.09)	0.07 (0.94)	0.01
Impacts_5–8 g_ (n/min)	1.7 ± 1.31 (0.14; 4.06)	1.77 ± 1.93 (0; 6.85)	−0.15 (0.88)	0.15
Acc_4–5 m/s_ (n/min)	0.01 ± 0.01 (0.14; 4.06)	0 ± 0.01 (0; 0.03)	1.04 (0.3)	0.14
Dec_6–5 m/s_ (n/min)	0 ± 0.01 (0; 0.03)	0 ± 0 (0; 0.02)	1.4 (0.17)	0.2
Acc_Max_ (m/s)	3.73 ± 1.06 (2.52; 7.43)	3.02 ± 0.82 (1.07; 4.29)	2.68 (0.01) *	0.4
Acc_6–10 m/s_ (n/min)	0 ± 0 (0; 0.02)	0 ± 0 (0; 0)	1.13 (0.26)	0.16
Acc_5–6 m/s_ (n/min)	0 ± 0 (0; 0.02)	0 ± 0 (0; 0)	1.13 (0.26)	0.16
TakeOff_5–8 g_ (n/min)	0 ± 0 (0; 0)	0 ± 0.01 (0; 0.03)	−1 (0.32)	0.14

* Significant differences with low effect size.

## Data Availability

The data used is available in the following link: 10.5281/zenodo.4616371.

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
