# Peer review of "Data Mining to Select Relevant Variables Influencing External and Internal Workload of Elite Blind 5-a-Side Soccer"

_ijerph, 2021, doi:10.3390/ijerph18063155_

Round 1
Reviewer 1 Report
The article is interesting and brings information about an area that is still very little explored in the literature. In the abstract I see the need to adjust the background, since this section was used to present only the purpose of the study. At the end of the introduction, three components are mentioned for obtaining the PCA, however the authors used 4. Wouldn't it be an expected conclusion that the heart rate of younger athletes would be higher? In addition, why this indicator was considered since it is a limited variable for describing the internal load of efforts in intermittent sports. I suggest checking the possibility of obtaining informations about environmental conditions. This would allow obtaining the UCTI score, which guarantees the conditions of environmental stress for the practice of sports. Most likely, the organization of the event has such information.Author Response
Dear Editor and reviewers:
We have carefully considered all reviewers' recommendations of the paper ID (ijerph-1126385) entitled "Data mining to select relevant variables influencing external and internal workload of elite blind 5-a-side soccer”. Please find enclosed our detailed answers to reviewers' queries. The authors declare that the manuscript is original and has not been considered for publication elsewhere. Additionally, the authors had approved the paper for release and are in agreement with its content.
Please find all corrections in red inside the manuscript.
Reviewer 1
R1.1. The article is interesting and brings information about an area that is still very little explored in the literature.
R/ Thank you for the opportunity to improve the final version of this manuscript.
In the abstract I see the need to adjust the background, since this section was used to present only the purpose of the study.
R/ Thank you for highlight this issue, we have added a background section to the abstract.
At the end of the introduction, three components are mentioned for obtaining the PCA, however the authors used 4.
R/ Thank you for highlight this issue, it was corrected and age was added as a component of the study.
Wouldn't it be an expected conclusion that the heart rate of younger athletes would be higher? In addition, why this indicator was considered since it is a limited variable for describing the internal load of efforts in intermittent sports.
R/ thank you for the support to improve the conclusions of this study. We have review and add this conclusions to the corresponding section. HR was considered as the most internal load variable studied in soccer research (Carling et al., Rampini et al., Nakamura et al., etc).
I suggest checking the possibility of obtaining informations about environmental conditions. This would allow obtaining the UCTI score, which guarantees the conditions of environmental stress for the practice of sports. Most likely, the organization of the event has such information.
R/ We have add the information of environmental conditions of the tournament as recommended. We really appreciate this suggestion.
Reviewer 2 Report
The article has merit. However, minor revisions must be done before acceptance [see Guide for Authors].
Minor spell check required. Furthermore, in this paper is necessary to improve some aspects.
- In abstract, it’s necessary to improve the writing enumeration of contextual indicators, because isn´t clear.
- In Keywords, it´s necessary to delete the numbers.
- In section 3, It’s necessary:
- to replace the values of the effect size (are always positives);
- to improve the write this section, because the authors only show tables with values.
- In Sections 4, 6 and 7, it is necessary to improve the writing.
- References should be described as follows in “Instructions for Authors” (https://www.mdpi.com/journal/ijerph/instructions).
Author Response
Dear Editor and reviewers:
We have carefully considered all reviewers' recommendations of the paper ID (ijerph-1126385) entitled "Data mining to select relevant variables influencing external and internal workload of elite blind 5-a-side soccer”. Please find enclosed our detailed answers to reviewers' queries. The authors declare that the manuscript is original and has not been considered for publication elsewhere. Additionally, the authors had approved the paper for release and are in agreement with its content.
Please find all corrections in red inside the manuscript.
Reviewer 2
Minor spell check required.
R/ We have corrected some minor/moderate spell issues, please see corrections throughout the manuscript.
Furthermore, in this paper is necessary to improve some aspects.
- In abstract, it’s necessary to improve the writing enumeration of contextual indicators, because isn´t clear.
R/Thank you for the opportunity to correct this issue. We have correct the citation of the contextual variables in the abstract.
- In Keywords, it´s necessary to delete the numbers.
R/ Thank you for highlight this issue. All numbers in keywords were deleted as requested by the reviewer.
- In section 3, It’s necessary:
- to replace the values of the effect size (are always positives);
R/ Thank you for pointing out this issue. We have corrected all the ES.
- to improve the write this section, because the authors only show tables with values.
R/ We really appreciate the opportunity to improve this section of the manuscript. Authors have added more interpretation to the results tables. We have added interpretation of the effect sizes values, referring to those variables that presented statistical differences.
- In Sections 4, 6 and 7, it is necessary to improve the writing.
R/ We are agree with the reviewer. Consequently, we have improve the writing of the sections as recommended.
References should be described as follows in “Instructions for Authors” (https://www.mdpi.com/journal/ijerph/instructions).
R/ Thank you for pointing out this issue, authors have corrected references sections as recommended based on journal guidelines.
Reviewer 3 Report
Thank you for giving me the possibility of reviewing this paper. I hope the authors find my comments productive and that it will help them to improve their research work.
In this article, the authors explore and differentiate those
significant variables of internal and external load in blind or disabled elite players visual impairment of soccer at 5 (Fa5).
Electronic Performance Analysis Devices (EPTS) are changing the sports sector, so research should be done in order to be aware of its benefits for sports and get their maximum utility.
The selection of the principal component analysis (PCA) method to reduce the amount of data extracted through data mining process, seems to me the most appropriate option as it is an investigation in which the variables have high rates of correlation.
To measure internal loads, I would have proposed the authors to combine the method of heart rate (HR) together with other techniques such as subjective perception of effort (PSE), oxygen consumption (VO) and blood lactate concentration (LAC).
Therefore, I recommend that the authors read and cite the following article “Pérez, R. I., López, G. H., Meroño, A. J. (2020). Quantification of internal training load in players futsal professionals. SPORT TK-EuroAmerican Journal of Sports Sciences, 75-86 to reinforce the theoretical framework of the research.
About the data mining process it is important to justify and refer properly to this process. In order to do so, i suggest the authors to read and refer to the following work Reyes-Menendez, A., Saura, J. R., & Filipe, F. (2020). Marketing challenges in the# MeToo era: Gaining business insights using an exploratory sentiment analysis. Heliyon, 6(3), e03626.
Regarding the measurement of external loads, the device used is perfect (WIMU, RealTrack Systems). Right now, it is one of the most innovative methods to extract external variables.
Perhaps, I would have asked the authors to elaborate a little more on what this model, as well as mention its main characteristics as it is one of the
methodologies used for research.
One of the characteristics of this inertial device to consider and develop would be the Integrated GPS.
For a further explanation I recommend reading the recent article Gómez-Carmona, C. D., Gamonales Puerto, J. M., Feu Molina, S.; Ibáñez, S. J. (2019). Study of the internal load and external through different instruments: a case study in formative soccer. Sportis, 5 (3), 444-468.
In general, the methods used are correct as well as the presentation of the analysis and its results. However, I believe that the article could not be published as it suffers from a structure of a scientific article. It remains to expose the theoretical framework of the research as well as the hypotheses to be developed.
An interesting topic to investigate would be to be able to predict the performance of Fa5 as well as future injuries through other technologies such as Machine learning.
Author Response
Dear Editor and reviewers:
We have carefully considered all reviewers' recommendations of the paper ID (ijerph-1126385) entitled "Data mining to select relevant variables influencing external and internal workload of elite blind 5-a-side soccer”. Please find enclosed our detailed answers to reviewers' queries. The authors declare that the manuscript is original and has not been considered for publication elsewhere. Additionally, the authors had approved the paper for release and are in agreement with its content.
Please find all corrections in red inside the manuscript.
Reviewer 3
In this article, the authors explore and differentiate those
significant variables of internal and external load in blind or disabled elite players visual impairment of soccer at 5 (Fa5).
R/ Thank you for the opportunity to improve the final version of this manuscript.
Electronic Performance Analysis Devices (EPTS) are changing the sports sector, so research should be done in order to be aware of its benefits for sports and get their maximum utility.
The selection of the principal component analysis (PCA) method to reduce the amount of data extracted through data mining process, seems to me the most appropriate option as it is an investigation in which the variables have high rates of correlation.
R/ Thank you for your insights in this sense, we have strengthen the basis of the application of PCA in performance analysis in sport. We have add a couple of reference than support this analysis.
To measure internal loads, I would have proposed the authors to combine the method of heart rate (HR) together with other techniques such as subjective perception of effort (PSE), oxygen consumption (VO) and blood lactate concentration (LAC).
R/ We really appreciate the suggestion to include the interaction of internal and biochemical-physiological variables to better understand the team behavior. We have taken note to future interventions in similar population. We have add the citation that you suggested.
Therefore, I recommend that the authors read and cite the following article “Pérez, R. I., López, G. H., Meroño, A. J. (2020). Quantification of internal training load in players futsal professionals. SPORT TK-EuroAmerican Journal of Sports Sciences, 75-86 to reinforce the theoretical framework of the research.
R/Thank you for the suggestion, we have added this citation as recommended.
About the data mining process it is important to justify and refer properly to this process. In order to do so, i suggest the authors to read and refer to the following work Reyes-Menendez, A., Saura, J. R., & Filipe, F. (2020). Marketing challenges in the# MeToo era: Gaining business insights using an exploratory sentiment analysis. Heliyon, 6(3), e03626.
R/Thank you for the suggestion, we have added this citation as recommended.
Regarding the measurement of external loads, the device used is perfect (WIMU, RealTrack Systems). Right now, it is one of the most innovative methods to extract external variables.
Perhaps, I would have asked the authors to elaborate a little more on what this model, as well as mention its main characteristics as it is one of the
methodologies used for research.
R/ Thank you very much! We have considered your suggestion and the description of the device was widely strengthened.
One of the characteristics of this inertial device to consider and develop would be the Integrated GPS.
For a further explanation I recommend reading the recent article Gómez-Carmona, C. D., Gamonales Puerto, J. M., Feu Molina, S.; Ibáñez, S. J. (2019). Study of the internal load and external through different instruments: a case study in formative soccer. Sportis, 5 (3), 444-468.
R/Thank you for the suggestion, we have added this citation as recommended.
In general, the methods used are correct as well as the presentation of the analysis and its results. However, I believe that the article could not be published as it suffers from a structure of a scientific article. It remains to expose the theoretical framework of the research as well as the hypotheses to be developed.
R/ Thank you very much for highlight this issue. As suggested by the reviewer, we have incorporated the hypotheses of the authors in relation to this study and the objectives set into the introduction section.
An interesting topic to investigate would be to be able to predict the performance of Fa5 as well as future injuries through other technologies such as Machine learning.
R/We really appreciate the opportunity to improve the final version of the manuscript considering your recommendations.
Round 2
Reviewer 3 Report
Dear authors, after the review i consider that my comments were properly addressed so that this paper can be considered for publication in current form